# Gut Bacterial Dysbiosis in Irritable Bowel Syndrome: a Case-Control Study and a Cross-Cohort Analysis Using Publicly Available Data Sets

Gun-Ha Kim,[a] Kihyun Lee,[b] Jung Ok Shim[c]

[a]Division of Pediatric Neurology, Department of Pediatrics, Korea Cancer Center Hospital, Korea Institute of Radiological & Medical Sciences, Seoul, South Korea
[b]Chunlab, Inc., Seoul, South Korea
[c]Division of Pediatric Gastroenterology, Hepatology, and Nutrition, Department of Pediatrics, Korea University College of Medicine, Seoul, South Korea

**ABSTRACT** Research on the gut microbiota in irritable bowel syndrome (IBS) shows discordant results due to inconsistent study designs or small sample sizes. This study aimed to characterize how gut microbiota in IBS patients differs from that in healthy controls by performing a case-control study and cross- and mega-cohort analysis. Multiple publicly shared data sets were examined by using a unified analytical approach. We performed 16S rRNA gene (V3-4) sequencing and taxonomic profiling of the gut bacterial communities. Fecal samples from children with IBS ($n = 19$) and age-matched healthy controls ($n = 24$) were used. Next, we analyzed 10 separate data sets using a unified data-processing and analytical approach. In total, 567 IBS patients and 487 healthy controls were examined. In our data sets, no significant differences existed in stool $\alpha$-diversity between IBS patients and healthy controls. After combining all the data sets using a unified data-processing method, we found significantly lower $\alpha$-diversity in IBS patients than in healthy controls. In addition, the relative abundance of 21 bacterial species differed between the IBS patients and healthy participants. Although the causal relationship is uncertain, gut bacterial dysbiosis is associated with IBS. Further functional studies are needed to assess whether the change in gut microorganisms contributes to the development of IBS.

**IMPORTANCE** Research on the gut bacteria in irritable bowel syndrome (IBS) shows discordant results due to inconsistent study designs or small sample sizes. To overcome these issues, we analyzed microbiota of 567 IBS patients and 487 healthy people from 10 shared data sets using a unified method. We demonstrated that gut bacteria are less diverse in IBS patients than in healthy people. In addition, the abundance of 21 bacterial species is different between the two groups. Altered bacterial balance, called dysbiosis, has been reported in several disease states. Although the causal relationship is uncertain, gut bacterial dysbiosis also seems to be associated with IBS.

**KEYWORDS** gastrointestinal, microbiota, 16S rRNA, diversity, microbiota-gut-brain axis, dysbiosis

Address correspondence to Jung Ok Shim, shimjo@korea.ac.kr.

The authors declare a conflict of interest. This work was supported by The National Research Foundation of Korea Grant, funded by the Korean government (Ministry of Science and ICT) (No. NRF-2018R1C1B5047245) and by Korea University Hospital Hin Moe (Hyun-Gum Lee) Research Fund.

Irritable bowel syndrome (IBS) is a common functional gastrointestinal disorder. However, the pathogenesis is unknown, and no effective treatment strategy exists yet. Based on the epidemiological studies of IBS patients, altered gut microbiota was proposed as one of the possible causes of IBS. Acute bacterial gastroenteritis can cause chronic, asymptomatic, low-grade intestinal wall inflammation sufficient to alter neuromuscular and epithelial cell function (1). In 36% of patients infected with *Campylobacter jejuni* and *Escherichia coli* O157:H7, IBS presented itself with heightened intestinal permeability, which was observed even after 2 years postinfection (2). Thus, it has been suggested that changes to the gut microbial community may trigger IBS symptoms. The importance of gut microbiota in the etiology of IBS

**TABLE 1** Comparison of the study population[a]

| Variables | D+ IBS (n = 10) | D− IBS (n = 9) | P value | Total IBS (n = 19) | HC (n = 24) | P value |
|---|---|---|---|---|---|---|
| Age, y | 13.1 ± 2.1 | 13.7 ± 1.9 | 0.875 | 13.4 ± 2.0 | 13.8 ± 2.2 | 0.577 |
| Sex, M:F | 6:4 | 5:4 | 1.000 | 11:8 | 17:7 | 0.574 |
| Duration of morbidity, y | 0.8 [0.2; 2.0] | 0.5 [0.2; 3.0] | 0.901 | 0.5 [0.2; 2.0] | NA | NA |
| Less than 1 yr/more than 1 yr | 5/5 | 5/4 | 1.000 | 10/9 | NA | NA |
| Calprotectin (mg/kg) | 12.0 [9.6;31.5] | 37.5 [8.1;71.0][b] | 0.197 | 26.5 [8.1;53.0] | NA | NA |
| <50/≥50 | 6/4 | 7/1 | 0.444 | 13/5 | NA | NA |

[a]The data are presented as means ± standard deviations for normally distributed continuous variables or medians with [interquartile range] for non-normally distributed variables. D+ IBS, Irritable bowel syndrome with diarrhea; D− IBS, irritable bowel syndrome without diarrhea; HC, healthy control; M, male; F, female; NA, not available.
[b]The stool calprotectin measurement was missing in one patient.

becomes more apparent after special diets and antibiotics have improved IBS symptoms. However, most studies on the gut microbiota in IBS show discordant results, perhaps due to inconsistent study designs or small study populations (3–5). Thus, we aimed to characterize how gut microbiota in IBS patients differs from that in healthy people. We carried out a case-control study and analyzed cross- and mega-cohort data from multiple shared data sets, using a unified analytical approach.

## RESULTS

**Case-control study. (i) Comparison of the study population.** We enrolled 24 HCs and 19 IBS patients. Of the IBS patients, 10 had IBS with diarrhea, and 9 had IBS without diarrhea (Table 1). There were no differences in age and sex between the two groups (mean age of 13.4 ± 2.0 years). The median [interquartile range] fecal calprotectin level in the IBS group was 26.5 [8.1;53.0], with no observed differences according to IBS type.

**(ii) Differences in the gut bacterial diversities.** We compared the $\alpha$-diversity (species diversity within a single sample) between the following groups: IBS patients (n = 19) versus HCs (n = 24); HCs versus IBS patients with diarrhea (n = 10) versus those without diarrhea (n = 9); patients morbid above a year (n = 9) versus below a year (n = 10); and subjects with calprotectin levels above versus below 50 mg/kg (Fig. 1A to D). The only observed significant difference was in the phylogenetic diversity between IBS with diarrhea and without diarrhea groups. For this finding, the unadjusted P values were 0.044 on the t test and 0.045 on Wilcoxon rank-sum test (Fig. 1B).

**(iii) Differences in bacterial community compositions.** We also analyzed the $\beta$-diversity (between-group dissimilarities in bacterial community composition) between each pair of groups (Fig. 2). When using the operational taxonomic unit (OTU) tables, bacterial community compositions differed significantly between the IBS group versus the HC (P = 0.018) and exhibited a weak association in the IBS without diarrhea group versus the HC (P = 0.024). No difference between HC and IBS with diarrhea was found (P = 0.57). A similar pattern was observed when the same analyses were repeated using the ASV composition table. In other words, the IBD with diarrhea group did not diverge from HC (P = 0.43), while the IBD without diarrhea group diverged from HC with marginal significance (P = 0.056) (Fig. 3).

**(iv) Taxonomic biomarkers.** We searched for microbial biomarkers of IBS and found several families, genera, OTUs, and ASVs that were enriched in diarrheal and nondiarrheal IBS patients. Christensenellaceae was markedly depleted at the family level in nondiarrheal IBS (P = 0.01; 11-fold) but not in diarrheal IBS (P = 0.88). Of note, none of the enriched or depleted taxa in the IBS with diarrhea overlapped with those in the IBS without diarrhea. An exception was a single OTU that belonged to poorly Clostridium_g24 (Fig. 4), suggesting that the microbial features associated with two types of IBS are distinct from each other.

**Cross-cohort analysis. (i) $\alpha$-Diversity comparison within individual studies.** Only one adult data set displayed a difference in $\alpha$-diversity (P = 1.7E−6, false discovery rate [FDR] = 1.7E−5), wherein IBS patients showed a lower $\alpha$-diversity than HCs (Fig. 5). One pediatric data set showed a slightly higher $\alpha$-diversity in IBS; however, no significant difference in the FDR-adjusted P value was observed.

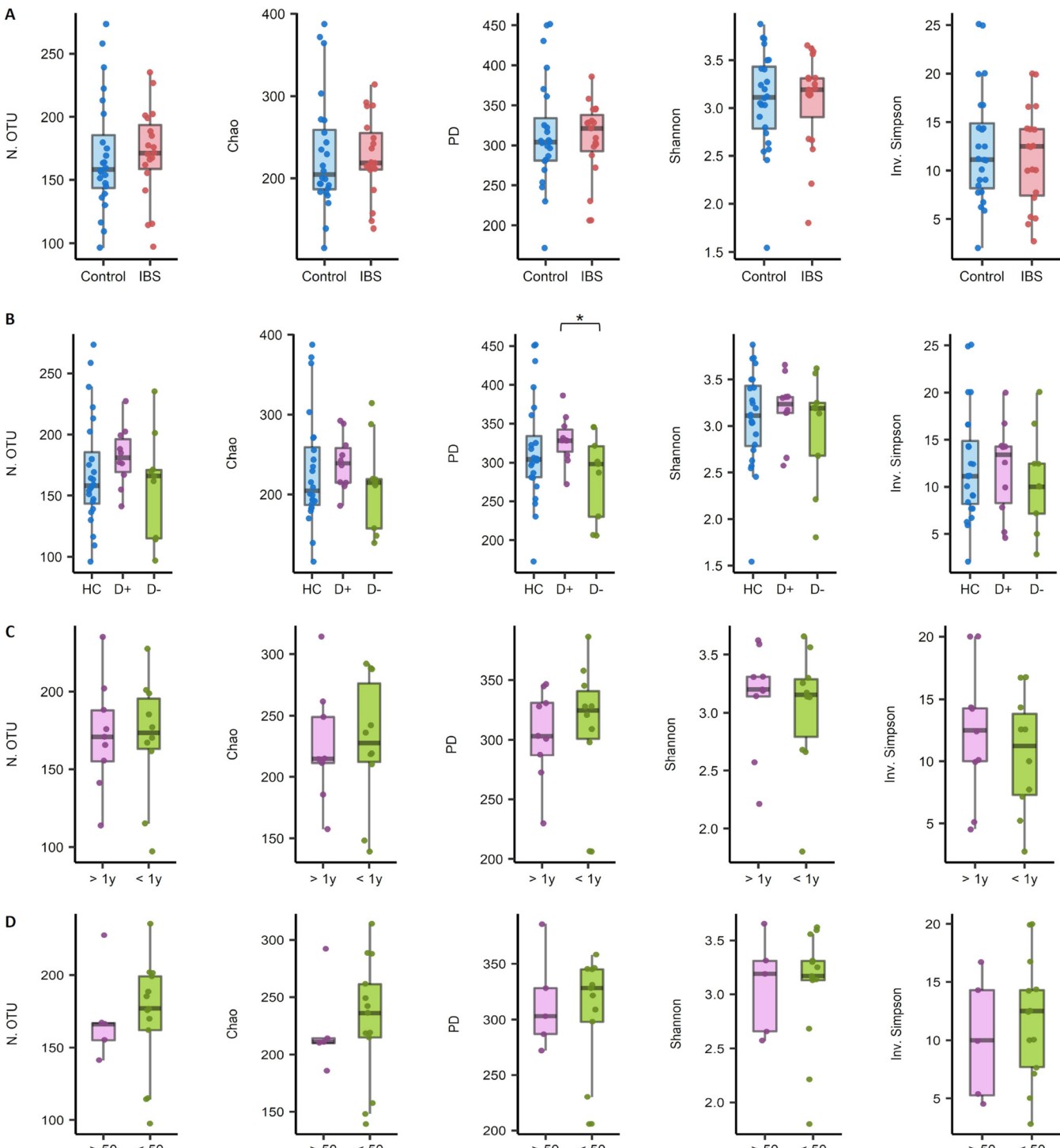

**FIG 1** Differences in the microbial $\alpha$-diversity indices in stool samples between groups. (A) Irritable bowel syndrome (IBS) patients ($n = 19$) versus healthy controls (HCs; $n = 24$). (B) HCs ($n = 24$) versus IBS patients with diarrhea ($n = 10$) versus those without diarrhea ($n = 9$). (C) Patients morbid above a year ($n = 9$) versus below a year ($n = 10$). (D) Calprotectin levels above ($n = 5$) versus below ($n = 13$) 50 mg/kg. unadjusted *, $P < 0.05$. N.OTU, number of operational taxonomic units; PD, phylogenetic diversity; Inv. Simpson, inverse Simpson's index; D+, IBS with diarrhea; D−, IBS without diarrhea.

In adults, the median Chao1 values were lower in IBS patients than in corresponding HCs (lower diversity; four of seven data sets). In contrast, these values for the pediatric IBS patients were higher than in the corresponding HCs (higher diversity; three of three data sets) (Fig. 5A). For the Shannon index (Fig. 5B), only the adult data set showed a difference in $\alpha$-diversity ($P = 2.9E-5$, FDR $= 2.9E-4$),

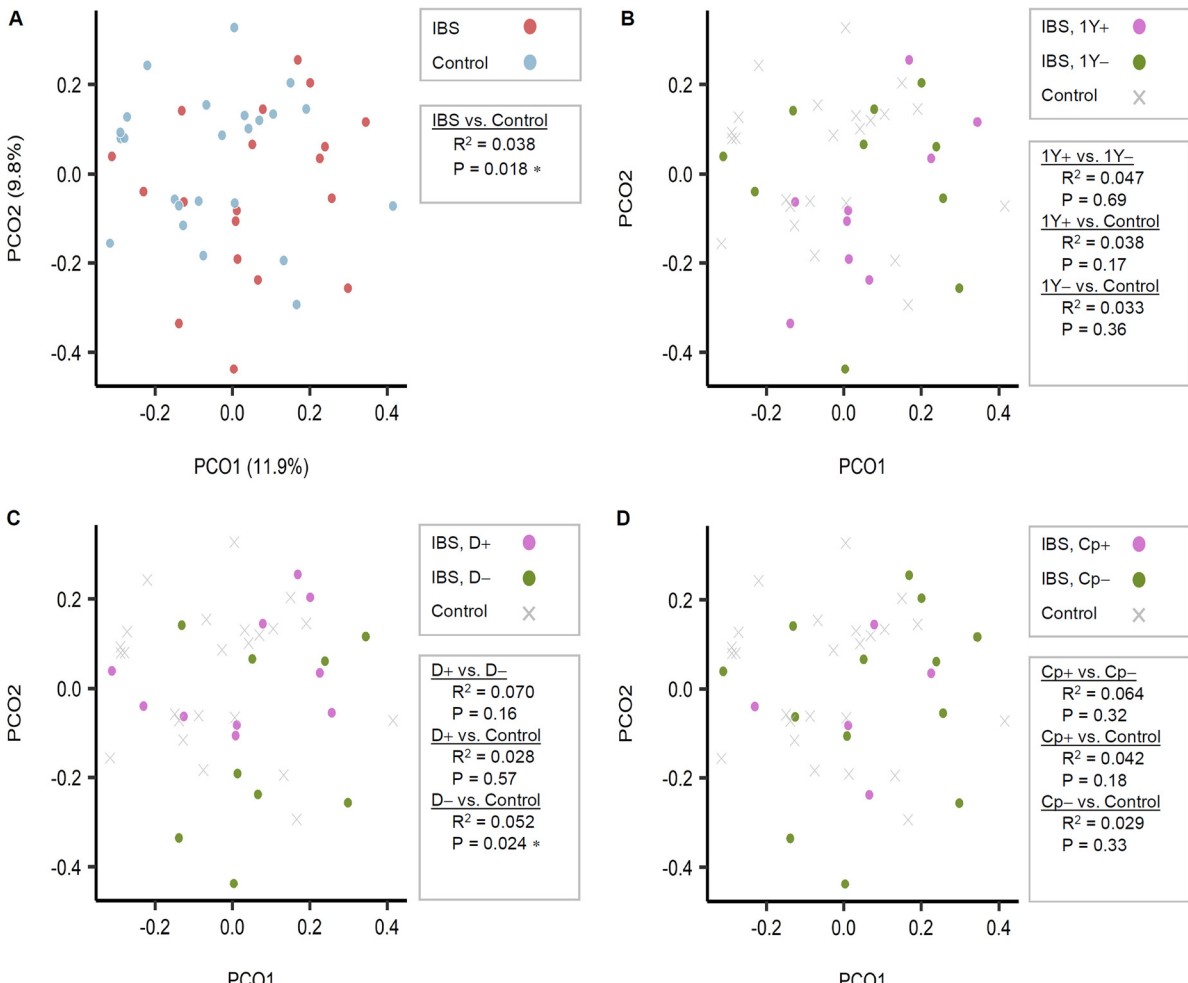

**FIG 2** Comparison of stool sample bacterial community compositions based on Bray-Curtis dissimilarity index among operational taxonomic unit (OTU) tables. (A) Irritable bowel syndrome (IBS) patients ($n = 19$) versus healthy controls ($n = 24$). (B) Patients morbid above a year ($n = 9$) versus those below a year ($n = 10$). (C) IBS patients with diarrhea ($n = 10$) versus those without diarrhea ($n = 9$). (D) Calprotectin levels ($n = 5$) versus below ($n = 13$) 50 mg/kg. PCO1, Principal coordinate 1; PCO2, Principal coordinate 2; N.OTU, number of OTU; D+, IBS with diarrhea; D−, IBS without diarrhea.

where IBS had a lower $\alpha$-diversity. No pediatric data set showed a difference in $\alpha$-diversity.

**(ii) $\beta$-Diversity comparison within individual studies.** We compared the gut microbial communities of each data set with the PCOA and Adonis tests using Aitchison distances (Fig. 6). Based on the OTU-level compositions (Fig. 6A), one of three pediatric data sets and four of seven adult data sets revealed differing community compositions between IBS and HC groups (Adonis test; $P < 0.05$). For the species-level compositions (Fig. 6B), two of three pediatric data sets and four of seven adult data sets had differing community compositions between IBS and HC. Regarding the genus-level compositions (Fig. 6C), two of three pediatric data sets and four of seven adult data sets had differing community compositions between IBS and HC. However, the correlation for it was low ($R^2 < 0.2$), except in the Zhu et al. 2019 data set (6) ($R^2 = 0.2$).

**(iii) Differentially abundant microbial taxa within individual studies.** We further analyzed the differentially abundant microbial taxa between IBS and HC for each data set using the ALDEx2 method (Fig. 7). Profiles were tested at the OTU, species, and genus levels. A taxon was considered differentially abundant between groups (green dots) if it satisfied one of the following criteria: (i) FDR < 0.1, (ii) effect size greater than 1 or less than −1, or (iii) the 95% confidence interval of the effect size does not span zero—i.e., the range is completely less than 0 or completely greater than 0. Notably,

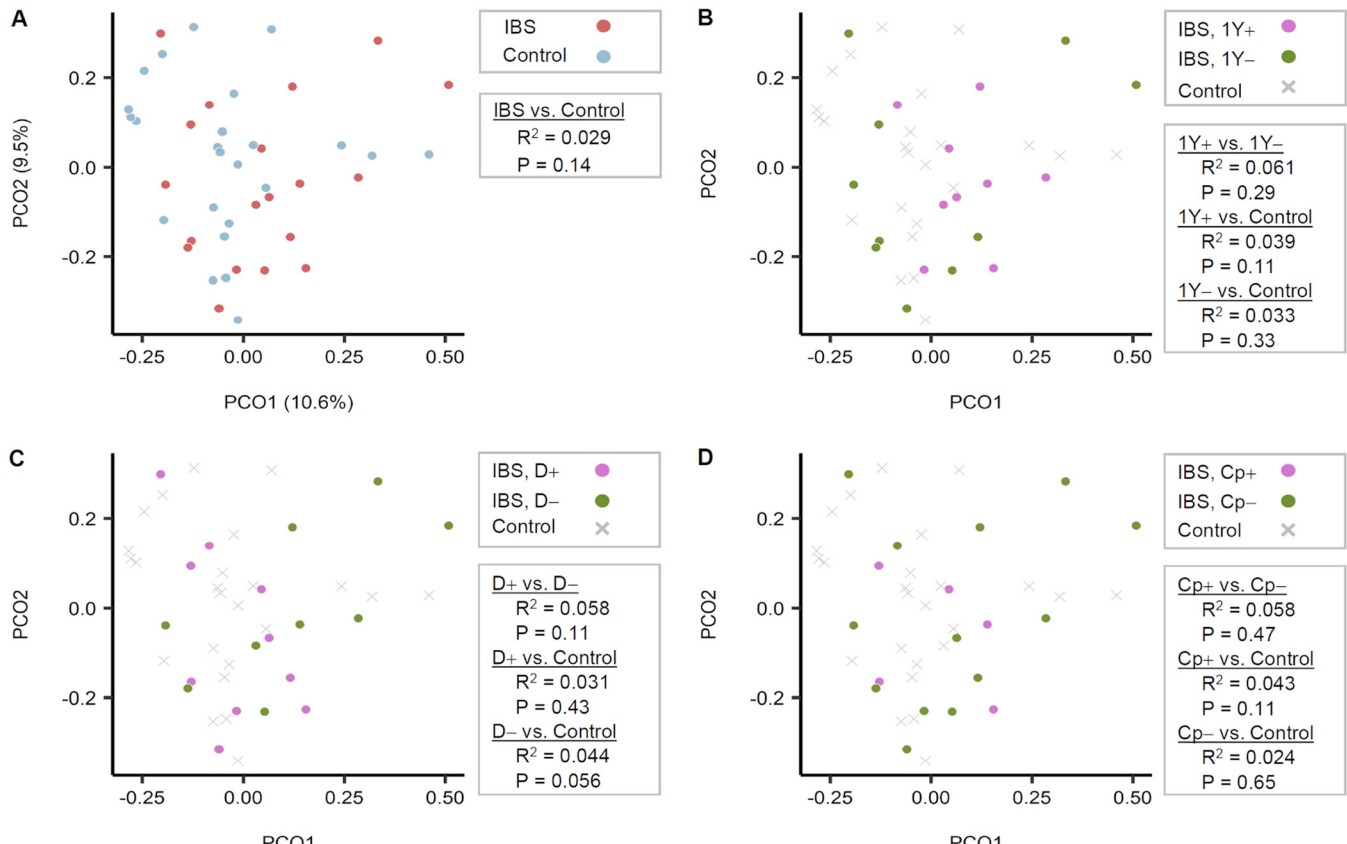

**FIG 3** Comparison of gut bacterial communities between groups based on the 16S rRNA gene amplicon sequence variants (ASVs). No differences were noted when using the 16S rRNA gene ASV pipeline. The Bray-Curtis dissimilarity matrix was calculated after rarefying the ASV table of each sample to have 7,500 reads/sample evenly. Principal coordinate analysis was performed on the Bray-Curtis dissimilarities, and the two major principal coordinates are displayed in the plots. Statistical significance of the separations between subject groups was tested pairwise with 999 permutations, using permutational multivariate analysis of variance (PERMANOVA) (Adonis). (A) Irritable bowel syndrome (IBS) patients (n = 19) versus healthy controls (n = 24). (B) Patients morbid above a year (n = 9) versus those below a year (n = 10). (C) IBS patients with diarrhea (n = 10) versus those without diarrhea (n = 9). (D) Calprotectin levels (n = 5) versus below (n = 13) 50 mg/kg. N.OTU, number of OTU; D+, IBS with diarrhea; D−, IBS without diarrhea; PCO1, Principal coordinate 1; PCO2, Principal coordinate 2.

the taxa that fit the criteria did not overlap in different studies: e.g., *Parasutterella* was enriched in the IBS group studied in the Lee 2021 data set (7); *Prevotella* was enriched in the IBS group analyzed in the Saulnier et al. 2011 data set (8); *Coprococcus*, *Oscillospiraceae* PAC000661_g, *Agathobaculum*, and *Ruminococcus* were depleted in the IBS group of the Pozuelo et al. 2015 study (9). The Zhu et al. 2019 data set (6) had too many taxa that passed the differential abundance criteria (purple dots). Thus, we determined the Zhu et al. 2019 data set (6) to be an outlier and excluded this data set from our analysis.

**Mega-cohort analysis: combining all data sets. (i) α-Diversity comparison.** The pediatric samples had significantly higher Chao1 α-diversity than adult samples (pediatric median 96.6 versus adult median 71.8, Wilcoxon test $P = 6.2E-42$; Fig. 8A). The α-diversity of all IBS patients was significantly lower than that of HC (IBS median 78.4 versus HC of all studies median 88.3, $P = 1.5E-8$; Fig. 8B) in adult data sets (adult IBS median 69.7 versus adult HC median 77.0, $P = 4.1E-6$; Fig. 8C). However, the pediatric data sets did not show a significant difference in α-diversity between children with IBS and HC (pediatric IBS median 96.6 versus pediatric HC median 96.6, Wilcoxon test $P = 0.58$; Fig. 8D). Next, we compared α-diversity between HC and IBS subtypes (with and without diarrhea), taking advantage of an increased sample size of the combined data set. This data set was comprised of 234 samples from HC (210 pediatric; 24 adults), 134 samples from IBS with diarrhea (10 pediatric; 124 adults), and 123 samples from IBS without diarrhea (9 pediatric; 114 adults). We excluded the studies in which

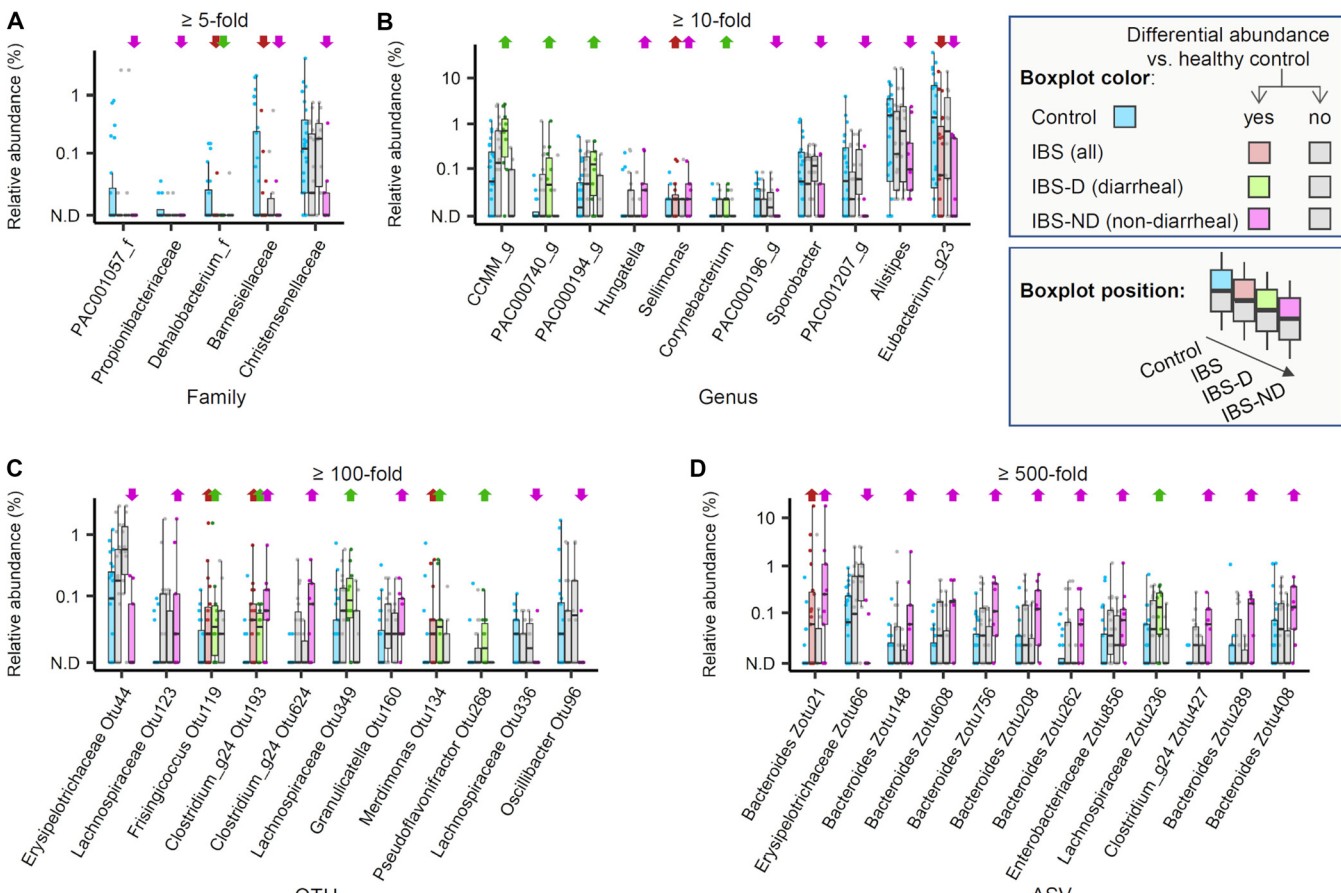

**FIG 4** Relative abundance of bacterial taxa in the stool samples obtained from irritable bowel syndrome patients and healthy controls. Four box plots are shown, representing the range of the relative abundance values of each taxon across individual samples in healthy control (HC), irritable bowel syndrome (IBS), diarrheal IBS (IBS-D), and nondiarrheal IBS (IBS-ND) subjects. The color scheme was used to aid distinction among the sample groups as well as the cases with significant differences (see the visual legend inside the figure). Significance was tested for a given taxon by performing $t$ test and Wilcoxon ranked sum test for comparisons between the healthy control group and each IBS category. A taxon was considered to be differentially abundant when the fold difference of abundance between the groups was above a certain threshold (5-fold for families, 10-fold for genera, 100-fold for operational taxonomic units [OTUs], and 500-fold for amplicon sequence variants [ASVs], adjusted for each rank to control the number of taxa to visualize), and $P < 0.05$ from $t$ test or Wilcoxon test. Upward and downward arrows are placed above the taxon in the given patient group. These indicate the relative enrichment and depletion, respectively, compared with abundance in HCs. Panel A-D show the selected families (A), genera (B), OTUs (C), and ASVs (D).

diarrheal subtype information was not provided (n samples excluded = 564). For Chao1, $\alpha$-diversity was lower in IBS with diarrhea than in HC; for the Shannon index, $\alpha$-diversity was lower in IBS with/without diarrhea than in HC (Fig. 8E). For pediatric patients, pooling samples across studies did not increase sample sizes in addition to what was provided in our cohort (Fig. 8F).

(ii) $\beta$-**Diversity comparison.** The species compositions were distinct between age groups ($R^2 = 0.048$, $P = 0.001$; Fig. 9A) and between IBS and HC ($R^2 = 0.0066$, $P = 0.001$; Fig. 9B), with a very weak correlation. In the pediatric data sets, the effect of data set origin (study-to-study) on species composition was apparent and significant ($R^2 = 0.18$, $P = 0.001$) and was stronger than the effect of disease state (HC/IBS) (Adonis $R^2 = 0.0057$, $P = 0.001$; Fig. 9C). Likewise, in the adult data sets, the effect of data set origin on the variation of species composition (Adonis, $R^2 = 0.14$, $P = 0.001$) was stronger than that of disease state (Adonis, $R^2 = 0.0045$, $P = 0.001$) or the IBS subtype (Adonis, $R^2 = 0.016$, $P = 0.001$; Fig. 9D). We used two adult cohorts, which covered both IBS patients with diarrhea (D+) and those without diarrhea (D−). Next, the effect of IBS subtypes on the species composition was tested. In the Altomare et al. 2021 data set (10), the divergence of microbiota composition from that of the HC was not observed for D+ patients ($P = 0.91$) but was marginally detected for D− patients ($P = 0.053$; Fig. 9E). In the Pozuelo et al. 2015 data set (9), the

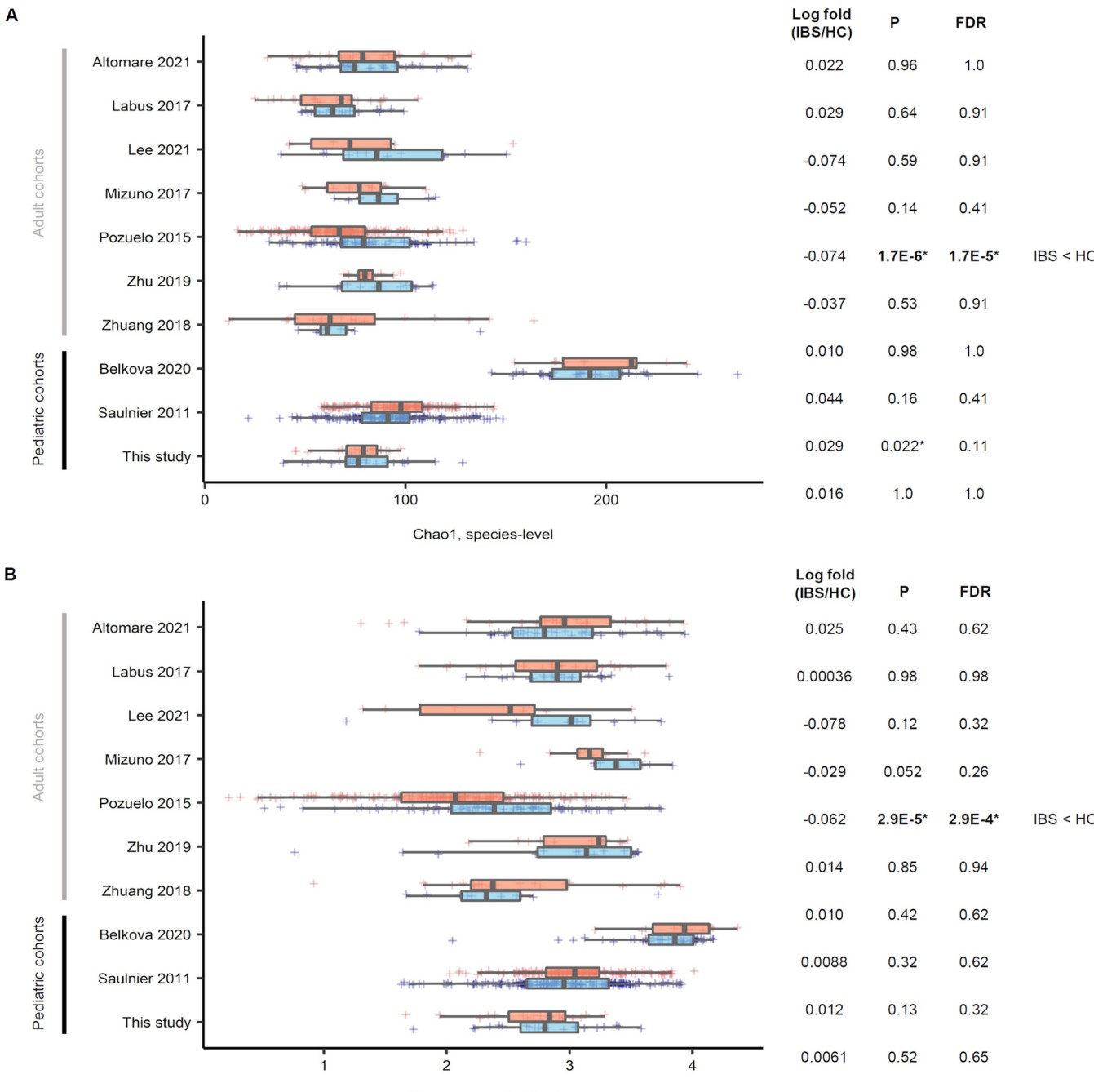

**FIG 5** Bacterial $\alpha$-diversity comparison between irritable bowel syndrome (all types) versus healthy control within 10 studies included in a cross-cohort analysis. For the Chao1 (A) and Shannon (B) index, only the Pozuelo et al. 2015 data set (9) showed a significant difference in $\alpha$-diversity ($P = 2.9E-5$, false discovery rate [FDR] = $2.9E-4$), where irritable bowel syndrome (IBS) had a lower $\alpha$-diversity across all data sets. *, $P < 0.05$. HC, healthy control.

divergence of microbiota composition from that of the HC was significant in both D+ and D− patients ($P = 0.001$ and $P = 0.006$, respectively).

**(iii) Differentially abundant species.** The $t$ test module of the ALDEx2 package was applied to three different sets: (i) all data sets combined, (ii) adult data sets, and (iii) pediatric data sets. The number of tested species in each set was 2,027 for all data sets combined, 1,553 for adult data sets, and 1,363 for pediatric data sets. Species with FDR < 0.1 were considered significant cases. In the combined data sets, 32 species were less abundant, while 6 were more abundant in IBS (Fig. 10). Across the three tests, 36 species were identified at least once as having a lower abundance in IBS, 10 of

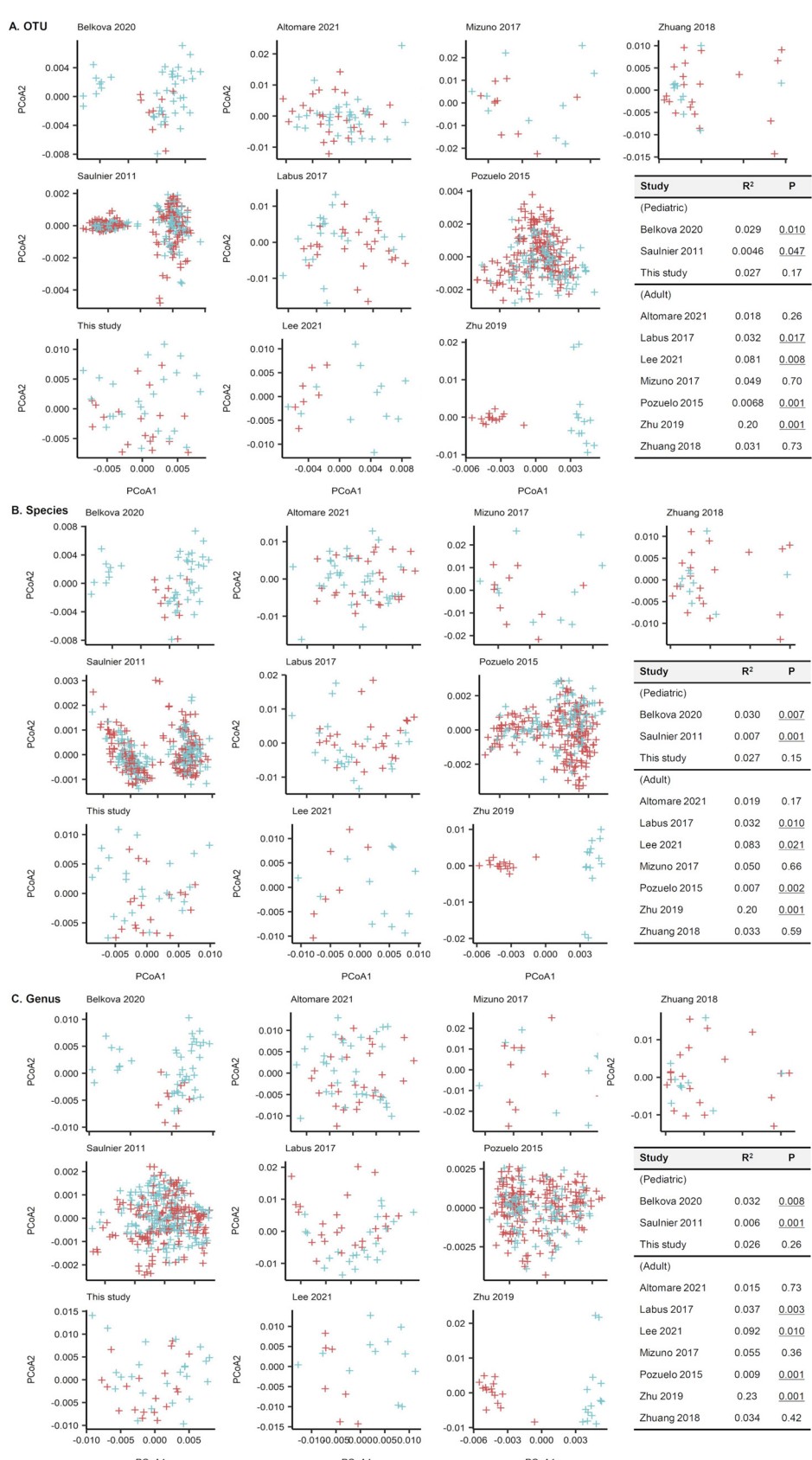

**FIG 6** Stool microbiota comparison of 10 cohorts using principle coordinates analysis and Adonis tests with Aitchison distances. (A) Operational taxonomic unit-level compositions. (B) Species-level compositions.

which were identified in more than one data set; 15 species were identified at least once as having a higher abundance in IBS, of which only 1 was identified in more than one data set; and 21 species (11 less abundant and 10 more abundant) were identified as differentially abundant in the pediatric data sets (Fig. 10D).

## DISCUSSION

In our case-control study, the $\alpha$-diversity of gut bacterial communities did not differ between children with IBS and HCs. Additionally, their species compositions were not distinguishable. However, a mega-cohort analysis combining our results with nine other data sets shared with BioProject accession numbers found that IBS patients had lower $\alpha$-diversity than HCs. In addition, 21 bacterial species showed differential abundance between IBS and HC stool samples. Despite this observation, the bacterial communities between IBS patients and HCs were poorly separated (significant but weak correlation, $R^2 = 0.0066$, $P = 0.001$; Fig. 9B). Instead, bacterial communities were more clearly separated according to each data set ($R^2 = 0.18$, $P = 0.001$; Fig. 9C).

Recently published meta-analyses only compared the relative abundance of specific bacterial species between IBS and HC patients (4, 11). On the other hand, we analyzed the abundance after combining 10 separate data sets through a unified data-processing and analytical approach. Our mega-cohort analysis encompassed 567 (360 adults, 207 children) IBS patients and 487 (243 adults, 244 children) HCs.

**Decreased $\alpha$-diversity in irritable bowel syndrome.** Our case-control study showed no overall difference in $\alpha$-diversity between groups, except for the phylogenetic diversity between IBS with and without diarrhea (Fig. 1B). In the cross-cohort study, four of seven adult data sets revealed a lower $\alpha$-diversity (Chao1) in IBS than in HC, of which only one data set showed statistical significance (Fig. 5A). After combining the data sets, we found that the $\alpha$-diversity was significantly lower in IBS with or without diarrhea than in adult HCs (Fig. 8C and E). However, it was not different between the two groups in the pediatric cohort (Fig. 8D and F). Our case-control study had a small sample size, and the other pediatric cohorts provided no subtype information.

Other studies have also reported a lower $\alpha$-diversity in IBS than in HC (9, 12–16), whereas some reported no difference in gut microbial diversity between the two groups (17, 18). A possible explanation for these contradictory findings might be an inconsistent study design (different data-processing and analytical methods) and the small sample sizes. Many variables other than IBS (e.g., food, sex, age, stress, drug, environment, or delivery methods) can affect the distribution of gut bacteria; thus, sample sizes need to be large enough to prove the hypothesis under highly variable conditions.

Although our results suggest that low gut microbial $\alpha$-diversity may predispose patients to IBS symptoms, low $\alpha$-diversity may result from a restricted diet low in fermentable oligosaccharides, disaccharides, monosaccharides, and polyols (FODMAP diet), which is often adopted to improve symptoms. The causality is uncertain because we do not have information about the patient food intake. However, one study found that decreased $\alpha$-diversity in IBS could be normalized closer to that of HC after 4 weeks on a low FODMAP diet (16).

**Heterogeneity of the included data sets.** We found that the bacterial communities of each data set were separated ($R^2 = 0.18$, $P = 0.001$; Fig. 9C). In addition to the differences in analytical methodology, different geographical origins and food cultures likely contributed to the different bacterial communities of each data set.

**Gut bacterial dysbiosis in irritable bowel syndrome.** We observed a lower gut bacterial diversity and a different gut bacterial species composition in IBS patients compared with those of HCs. The disruption of the physiologic symbiotic balance of

**FIG 6** Legend (Continued)

(C) Genus-name compositions. A few data sets showed significantly different distributions between patients with irritable bowel syndrome (IBS) and healthy controls (HC) ($P$ values that are below 0.05 are presented underlined in the inserted table). However, the correlation was low ($R^2 < 0.20$) except for the Zhu et al. 2019 data set (6) ($R^2 = 0.20$).

## A. ALDEx2 differentially abundant species

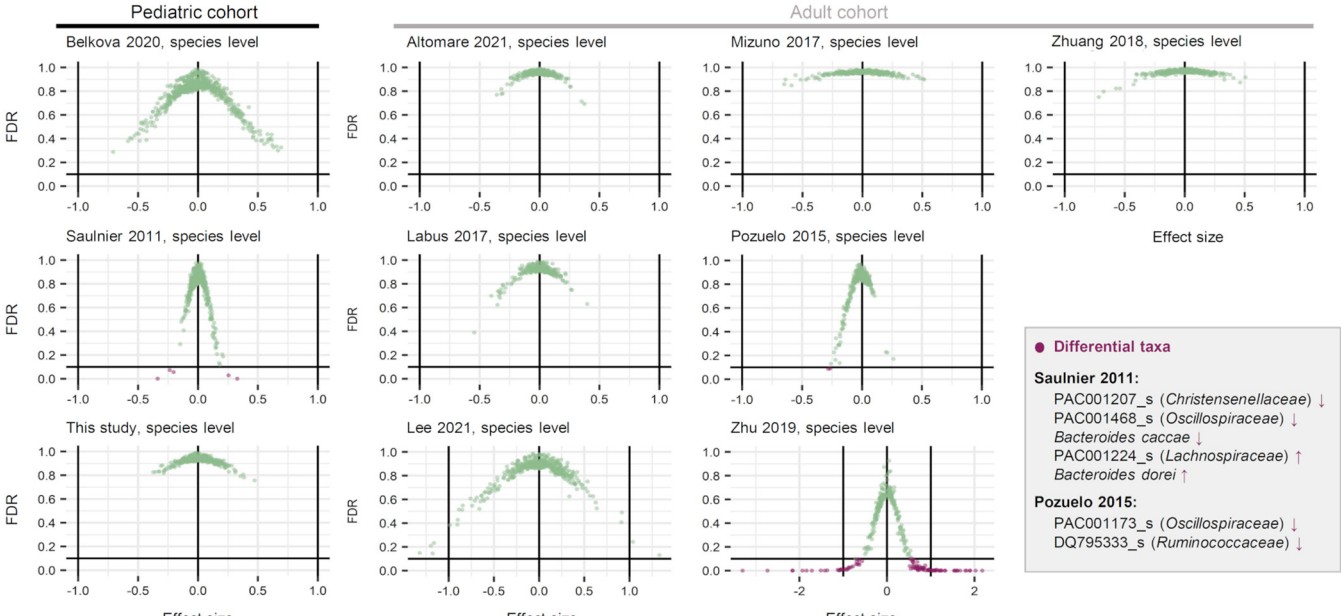

## B. ALDEx2 differentially abundant genera

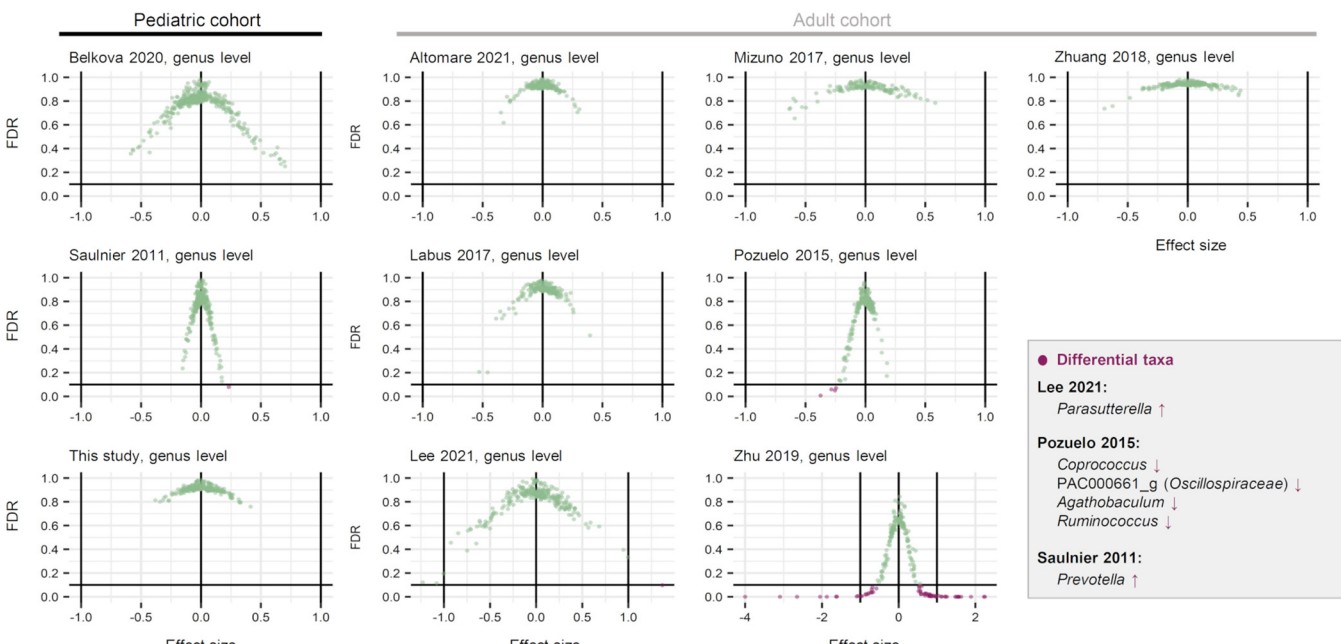

**FIG 7** Differentially abundant microbial taxa at the species (A) and genus (B) levels were determined using the ALDEx2 method. Each scatterplot displays the effective size (x axis) and false discovery rate (FDR; y axis) determined for each individual taxon. The data points located in the lower left area of the plot (i.e., those with FDR below 0.1 line and negative effective size) represent the taxa significantly depleted in IBS, while the data points located in the lower right area represent the taxa significantly enriched in IBS. The names of the taxa enriched or depleted in the IBS are shown in the text boxes. The Zhu et al. 2019 data set (6) had multiple taxa that passed the differential abundance criteria; Therefore, we concluded the data set to be an outlier and excluded it from the analysis.

the host and gut microbiota is called "dysbiosis" and is assumed to be an initiating factor in IBS. Gut bacterial dysbiosis is frequently seen in postinfectious IBS and is often comorbid with small intestinal bacterial overgrowth (19). Moreover, the eradication of small intestinal bacterial overgrowth reduces the symptoms of IBS (20). The beneficial effects of certain probiotics and nonsystemic antibiotics are also frequent findings in patients with IBS (2). The integrated neurohumoral communication between the microbiota, gut, and brain, known as the microbiota-gut-brain axis (MGBA), is central

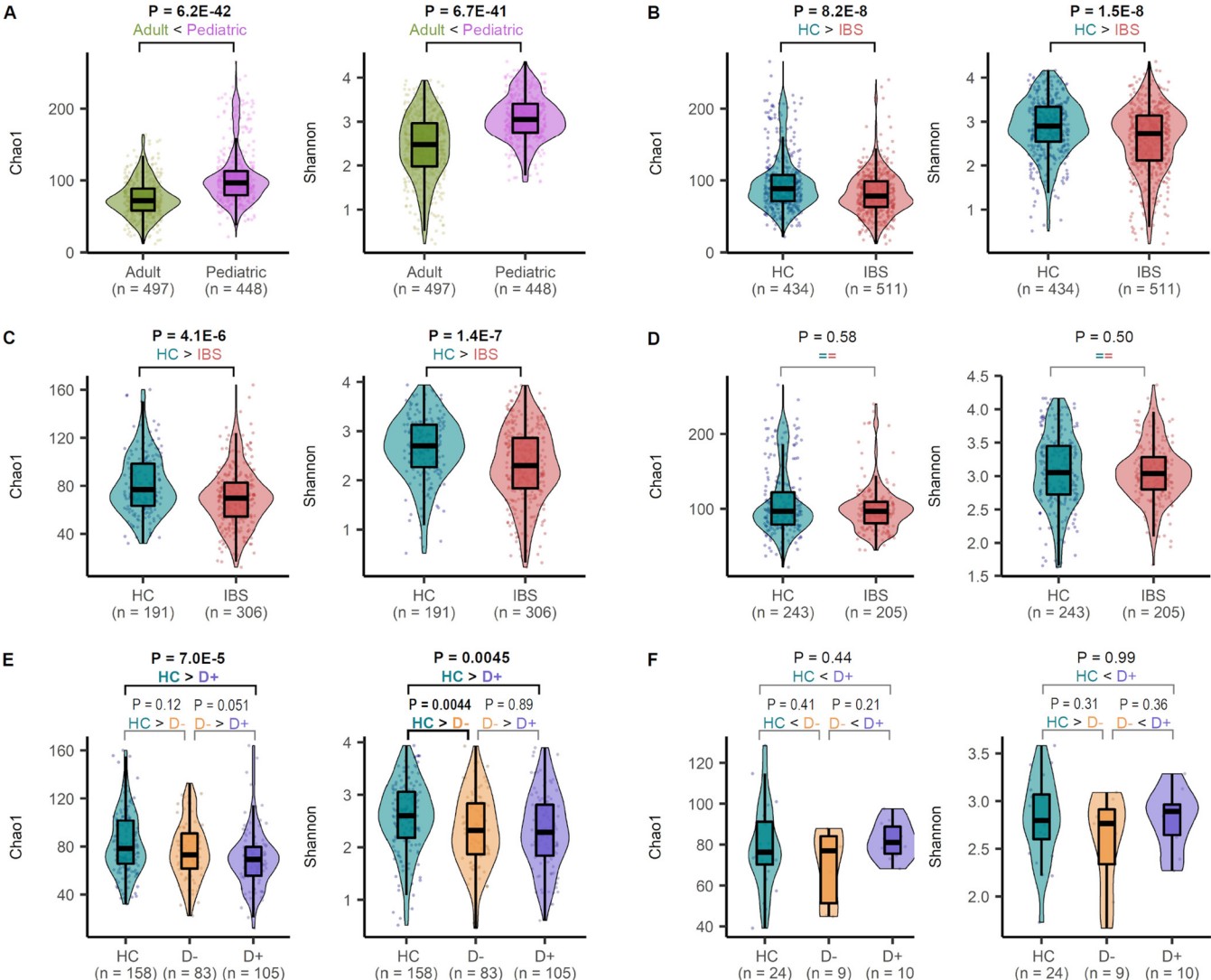

**FIG 8** Comparison of α-diversity in irritable bowel syndrome (IBS) patients and healthy controls (HCs). The box plots show the median and interquartile range of Chao1 index and Shannon index for subjects, as defined in the x axis. Individual data points and violin plots are shown to display the distribution of the values. The Chao1 and Shannon indices were calculated from the operational taxonomic unit tables after rarefying the number of reads to 1,000. P values determined by the Wilcoxon test are displayed over the groups in bold when P < 0.05. (A) Comparison between the pediatric and adult subjects after pooling the samples across all cohorts. (B) Comparison between HC and IBS patients. (C) Comparison between HC and IBS within the adult cohorts. (D) Comparison between HC and IBS within the pediatric cohorts. (E) Comparing HC, IBS with diarrhea, and IBS without diarrhea within the adult cohorts. (F) Comparison among HC, IBS with diarrhea, and IBS without diarrhea within the pediatric cohort. D+, IBS with diarrhea; D−, IBS without diarrhea.

to IBS symptoms. The MGBA modulates gastrointestinal functions such as intestinal motility and secretion, thus contributing to visceral hypersensitivity and cellular alterations of the entero-endocrine system (21).

**Limitations.** First, we combined data sets that used different regions of 16S rRNA genes and sequencing platforms. Second, a relatively short wash-out period was included in our case-control study. We recruited children with no use of antibiotics for at least 2 weeks prior to enrollment (visit 1) and collected stools 2 to 4 weeks after visit 1 with no use of antibiotics (visit 2). Therefore, the wash-out period without antibiotics was only 4 to 6 weeks. Last, stool samples, not mucosal samples, were used. It is known that stool microbiota differs from mucosal microbiota (22). The biofilm-like mucosal microbiota directly exchanges nutrients and induces the host's innate immunity; it is thus more influential to the MGBA function. Nevertheless, due to easy collection, stool samples are frequently used to investigate gut microbiota, allowing for large data sets.

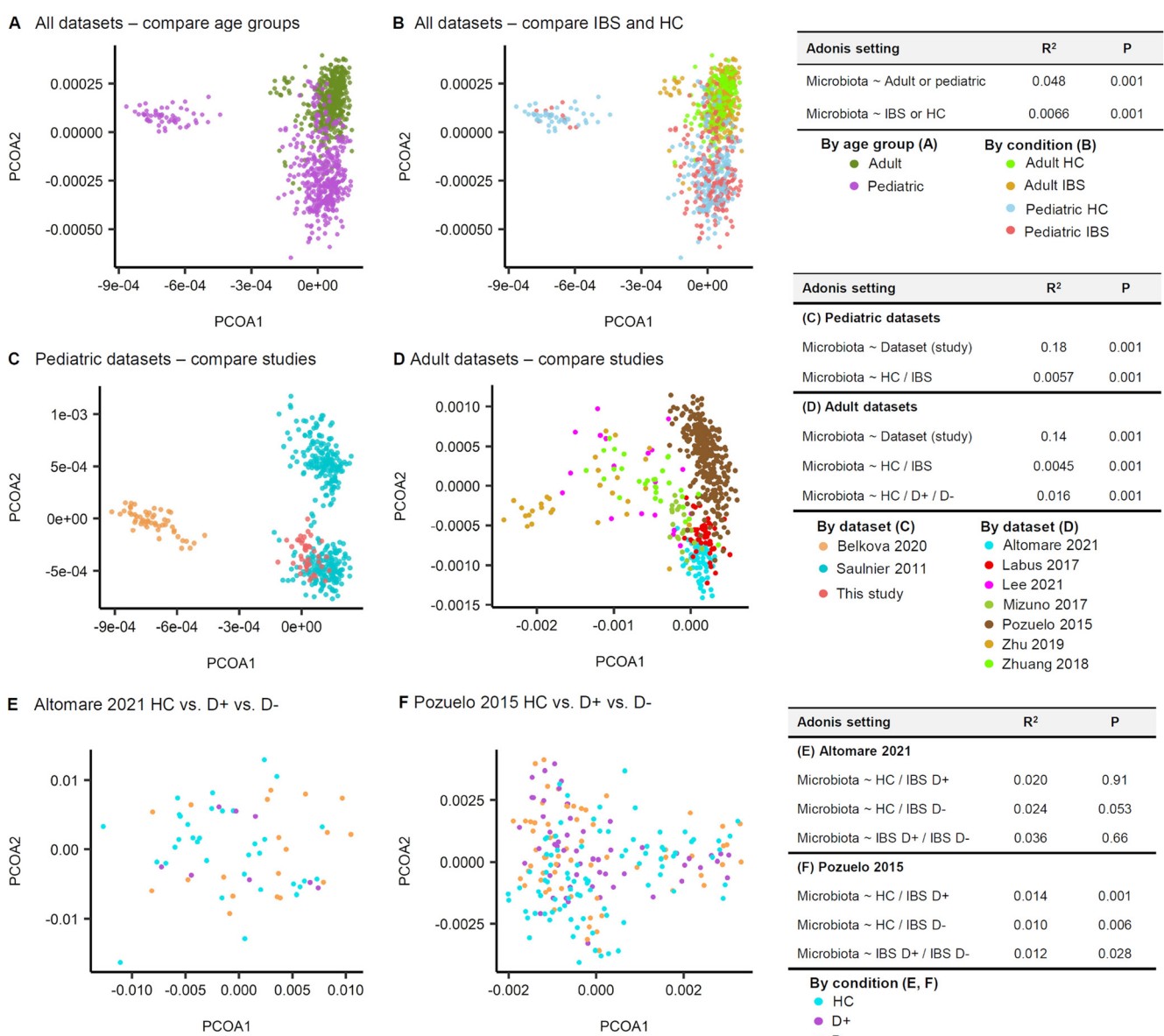

**FIG 9** Species compositions of irritable bowel syndrome (IBS) patients and healthy controls (HC). (A, B) Age group (A) was more determinant of species compositions than the disease state (B). (C) Based on the Adonis tests of the pediatric data sets, the effect of data set origin (study-to-study) on species composition was apparent and significant (Adonis, $R^2 = 0.18$, $P = 0.001$) and stronger than the effect of disease state (HC/IBS) (Adonis $R^2 = 0.0057$, $P = 0.001$). (D) Likewise, in the adult data sets, the effect of data set origin on the variation of species composition (Adonis, $R^2 = 0.14$, $P = 0.001$) was stronger than that of disease state (Adonis, $R^2 = 0.0045$, $P = 0.001$) or IBS subtype (Adonis, $R^2 = 0.016$, $P = 0.001$). Using the two adult cohorts, which included both the IBS patients with diarrhea (D+) and the IBS patients without diarrhea (D−), the effect of IBS subtypes on the species composition was tested. (E) In the Altomare et al. 2021 data set (10), the divergence of microbiota composition from that of the HC was not observed for D+ patients ($P = 0.91$), but marginally detected for D− patients ($P = 0.053$). (F) In the Pozuelo et al. 2015 data set (9), the divergence of microbiota composition from that of the HC was significant in both D+ and D− patients ($P = 0.001$, $P = 0.006$, respectively).

**Conclusions.** To our knowledge, we performed the first cross- and mega-cohort analysis to find the association between IBS and gut microbial diversity and composition. Our findings revealed that gut bacterial dysbiosis is associated with IBS, but the causal relationship is uncertain. Further studies are needed to ascertain whether the change in intestinal microorganisms contributes to developing IBS.

## MATERIALS AND METHODS

**Case-control study. (i) Study design and data collection.** We recruited children (4 to 19 years old) diagnosed with IBS and age-matched healthy participants from the Korea University Guro Hospital, Seoul, South Korea. The inclusion criteria for the patient group were as follows:

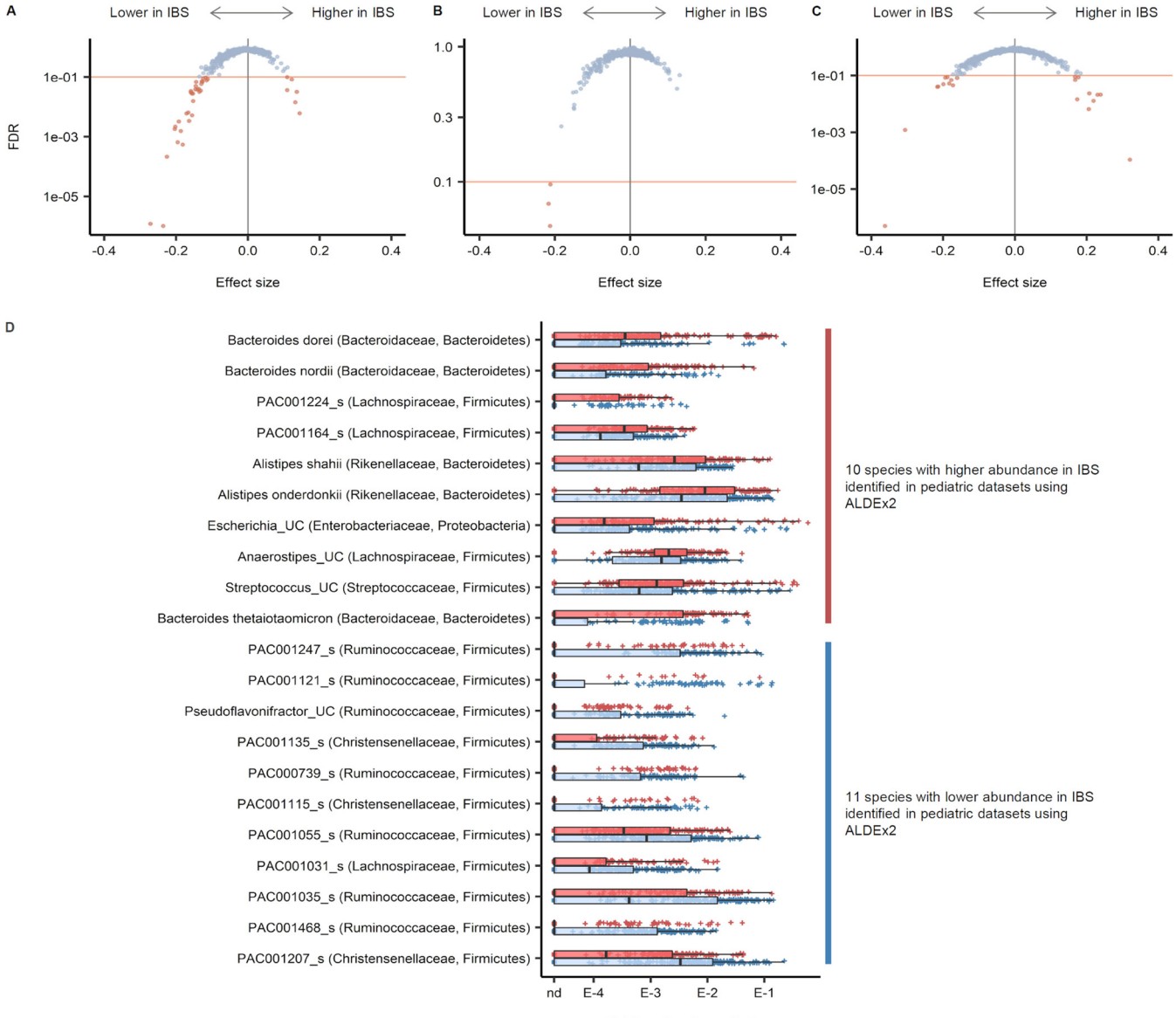

**FIG 10** Differentially abundant bacterial species in stool samples from irritable bowel syndrome (IBS) and healthy control (HC) patients. The *t* test module for the ALDEx2 package was applied to three different sets. (A) All data sets combined. (B) Adult data sets. (C) Pediatric data sets. (D) Visualization of the 21 species identified to be differentially abundant in pediatric data sets.

1.  Children diagnosed with IBS by the Rome IV criteria (H2b)
2.  No history of chronic diseases
3.  No other acute diseases within the previous 2 weeks
4.  No use of antibiotics or probiotics within 2 weeks prior to enrollment (stools were collected 2 to 4 weeks after the enrollment)
5.  Not obese (body mass index for age < 95 percentile)

We collected the clinical data on patient age, sex, age at IBS onset and diagnosis, disease type (i.e., diarrhea predominance versus with constipation or unspecified IBS), and fecal calprotectin levels. Age-matched children without any acute or chronic illness symptoms who did not consume probiotics or antibiotics within the previous 4 weeks of enrollment were recruited as the healthy controls (HCs). Children with high fecal calprotectin levels (>100 mg/kg) were subjected to an endoscopic evaluation to rule out other gastrointestinal disorders, such as inflammatory bowel disease, and to confirm the diagnosis of IBS.

**(ii) DNA extraction and 16S rRNA gene library preparation.** Fecal samples were frozen at −20°C until the DNA extraction was performed using a PowerSoil DNA isolation kit (Qiagen, CA, USA) following the manufacturer's instructions. The universal primer pair SD-Bact-0341-bS-17 and SD-Bact-0785-aA-21 were used for PCR amplification of the 16S rRNA gene V3-4 region. Purified amplicon libraries were sequenced using the Illumina MiSeq platform (Illumina, USA).

**TABLE 2** Summary of the studies included in the cross-cohort analysis[a]

| Reference | IBS/HC (no.), age group | DNA/RNA extraction kit or method | Region of 16S rRNA genes/ sequence platform |
|---|---|---|---|
| Altomare et al. (10) | 77/86, adults | QIAamp DNA stool minikit (Qiagen, Germany) | V1 to V3/454 Junior GS |
| Labus et al. (27) | 29/23, adults | MO BIO PowerSoil DNA isolation kit (MO BIO Laboratory, CA, USA) | V3 to V5/454 GS |
| Lee et al. (7) | 7/12, adults | QIAamp DNA stool minikits (Qiagen, Venlo, Netherlands) | V3 to V4/Illumina MiSeq |
| Mizuno et al. (28) | 10/10, adults | Enzymatic lysis method using lysozyme (Sigma-Aldrich Co. LCC, Tokyo, Japan) and achromopeptidase (Wako) | V1 and V2/Illumina MiSeq |
| Pozuelo et al. (9) | 202/88, adults | Qiaex II gel extraction kit (Qiagen, Hilden, Germany) | V4/Illumina MiSeq |
| Zhu et al. (6) | 15/14, adults | MO BIO PowerSoil DNA isolation kit (MO BIO Laboratory, CA, USA) | V4/Illumina HiSeq 2500 |
| Zhuang et al. (12) | 20/10, adults | PowerFecal DNA isolation kit (MoBio, Carlsbad, CA, USA) | V3 and V4/Illumina MiSeq |
| Belkova et al. (29) | 10/43, children | Zirconia beads (BeadBug, 0.5-mm zirconium beads, Sigma, USA) and QIAamp DNA stool kit (Qiagen, Germany) | V3 and V4/Illumina HiSeq |
| Saulnier et al. (8) | 178/177, children | MO BIO PowerSoil DNA isolation kit (MO BIO Laboratory, CA, USA) with modifications | V3 to V5/454 GS FLX |
| Our study | 19/24, children | MO BIO PowerSoil DNA isolation kit (MO BIO Laboratory, CA, USA) | V3 and V4/Illumina MiSeq |

[a]IBS, irritable bowel syndrome; HC, healthy control.

**Cross- and mega-cohort analysis data sets.** 16S rRNA gene sequence data of stool samples were obtained from previously published studies using their BioProject accession numbers. We collected data from ten studies (Table 2), which included seven adult and three pediatric data sets (including ours). In total, stool data from 567 IBS patients (360 adults and 207 children) and 487 HCs (243 adults and 244 children) were included in the cross- and mega-cohort analysis.

**Microbiota taxonomic profiles.** The paired-end sequencing reads generated in this study generated OTUs at a 97% sequence identity threshold. We taxonomically classified the OTUs using the EzBioCloud microbiome taxonomic profiling platform (23). Taxon names assigned by the EzBioCloud platform were based on the PKSSU4.0 database, which contains several modifications to maintain a consistent domain-to-species hierarchy. The complete taxonomic set used by the platform can be viewed at the EzBioCloud website (https://www.ezbiocloud.net/mtp/taxonomy).

Additionally, we generated amplicon sequence variants (ASVs) from the same sequence reads using the following steps. First, paired-end reads were quality-filtered and merged using fastp version 0.21.0 (24). Next, the merged reads were oriented, trimmed by 20 bp from both ends, and filtered with a maximum expected error cutoff of 1.0 using Usearch version 11 (25). Preprocessed reads were pooled and dereplicated into unique sequences using the fastx_uniques command. They were denoised into ASVs using the unoise3 command with a minimum size cutoff of five reads. Both commands were implemented in Usearch version 11.

In this study, the publicly available data sets that were reanalyzed displayed heterogeneity based on the sequencing technology used (i.e., pyrosequencing-based versus Illumina sequencing by synthesis), read length, amplicon target region, and sequencing layout (i.e., single-end versus paired-end). Due to such limitations, we applied an OTU-based approach only to the comparisons within each data set. To compare the samples from all data sets, we used genus and species profiles rather than the OTUs. The preprocessing steps described above were used for our paired-end sequences and were applied identically to the public data generated with Illumina platforms. The initial step using fastp was omitted for pyrosequencing data, and additional read length truncation was applied with the fastq_filter command of Usearch version 11. Furthermore, OTU clustering was performed with the cluster_otus command of Usearch version 11. The OTUs were taxonomically classified using the PKSSU4.0 reference database accessed through the EzBioCloud website (https://www.ezbiocloud.net/resources/16s_download).

**Microbiota comparison.** To compare within-data set microbial community $\alpha$-diversity, we rarefied the OTU table for each sample size within the data set. We evenly rarefied read count tables to 1,000 reads/sample for comparisons encompassing different data sets and discarded the samples with less than 1,000 reads. Rarefying was performed with the rarefy_even_depth command of the phyloseq package in R. $\alpha$-Diversity estimates were calculated for each subsampled read count table using the estimate_richness command of the phyloseq package. These calculations were repeated 100 times, and the median value for each diversity index was used as the final value.

We used either Bray-Curtis or Aitchison distances to assess the microbial community $\beta$-diversity. We calculated Bray-Curtis dissimilarities with the vegdist command of the Vegan R package using relative abundance tables (i.e., taxon read count divided by the total read count) as input. After performing a centered log-ratio transformation on the original read counts, using the clr command of the Compositions R package, we calculated Aitchison distances using the Euclidean method in vegdist. Compositional variation was visualized by principal coordinate analysis. Associations with the sample conditions (e.g., data sets, age, IBS) were also tested using permutational multivariate analysis of variance using the Adonis command of the Vegan package.

We tested for differential taxa abundance between groups by applying the Wilcoxon rank-sum test on the relative abundance distribution of each taxon or applying the ALDEx2 method to the taxa-sample read count table (26). We corrected the P values using the Benjamini-Hochberg method after carrying out the Wilcoxon rank-sum test on multiple taxa or data sets.

**Ethics approval and consent to participate.** We obtained written informed consent from all participants and their parents prior to enrollment in the study and sample collection. The Institutional Review Board of Korea University Guro Hospital approved this study (No. 2020GR0509).

**Data availability.** We have uploaded data files to zenodo with the DOI 10.5281/zenodo.7272051 (https://doi.org/10.5281/zenodo.7272051). Files uploaded include the following: (i) ASV sequences of all samples; (ii) ASV read count matrix of all samples; (iii) metadata table files, one per each data set, including those from our own and other published studies; (iv) combined all-in-one metadata table restricted to IBS versus control state; (v) OTU read count matrix for each data set; (vi) OTU taxonomy table for each data set; and (vii) species- and genus-level composition matrices for each data set.

## ACKNOWLEDGMENTS

We declare no conflict of interest.

This work was supported by a National Research Foundation of Korea grant, which was funded by the Korean government through Ministry of Science and ICT grant No. NRF-2018R1C1B5047245.

J.O.S. designed the study, gathered the data, and revised the manuscript; K.L. analyzed and interpreted the data; and G.-H.K. interpreted the data, drafted the work, and substantively revised it. All authors read and approved the final manuscript.

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
