## [Reviewer comments · Microbiology Spectrum]

Microbiology Spectrum

Gut bacterial dysbiosis in irritable bowel syndrome; a case-control study and a cross-cohort analysis of publicly available datasets

Gun-Ha Kim, Kihyun Lee, and Jung Ok Shim

Corresponding Author(s): Jung Ok Shim, Korea University College of Medicine

Review Timeline:

Submission Date:	June 23, 2022
Editorial Decision:	September 26, 2022
Revision Received:	November 23, 2022
Accepted:	December 10, 2022

Editor: Jonathan Jacobs

Reviewer(s): Disclosure of reviewer identity is with reference to reviewer comments included in decision letter(s). The following individuals involved in review of your submission have agreed to reveal their identity: Walaa K. Mousa (Reviewer #2)

Transaction Report:

DOI: <https://doi.org/10.1128/spectrum.02125-22>

September 26, 2022

Prof. Jung Ok Shim
Korea University College of Medicine
148, Gurodong-ro, Guro-gu
Seoul 08308
Korea (South), Republic of

Re: Spectrum02125-22 (Gut bacterial dysbiosis in irritable bowel syndrome; a case-control study and a cross-cohort analysis of publicly available datasets)

Dear Prof. Jung Ok Shim:

Your manuscript requires substantial revisions to meet the reviewers and my concerns about the robustness of the results. In addition, you may want to work with a professional editing service to improve the English readability of the manuscript, which could benefit from revisions. Please address all the reviewers concerns and reflect those changes in an updated manuscript. Furthermore, I strongly advise that claims made about the therapeutic or diagnostic potential of these results be revised to reflect the early preliminary observations of this report.

Link Not Available

Sincerely,

Jonathan Jacobs

Journals Department
Reviewer comments:

Reviewer #1 (Comments for the Author):

IBS is a significant functional GI disorder affecting many millions of people across the globe. What causes IBS remains elusive, but it is believed that gut microbiome may play a significant role. Here, the authors first conducted a small case-control study with a focus on associating IBS with alpha and beta bacterial diversity. It appeared that while the association with alpha diversity

was largely absent (Figure 1), it was significant with beta diversity when measured by OTUs (Figure 2). Notably, this significance only remained for the IBS without diarrhea but disappeared for the IBS with diarrhea (IBS-D) subtype. This suggested the IBS patients here maintained an equally rich but distinct microbiome (only IBS without diarrhea) compared to the health controls (HC). The same trend was also observed when measured by ASVs (Figure 3), offering robustness. This distinct microbiome was further illustrated by the differential relative abundance of certain bacterial taxa at the family, genera, species, and ASVs levels (Figure 4), but not the OTU levels. Because partial 16S does not provide species level resolution, the authors should replace the species plot with an OTU plot in Figure 4. Moreover, the authors should have a 4-box plot instead of the 2-box, i.e., IBS D+ vs IBS D- vs IBS vs HC.

Next, the same computational pipeline was applied to a cross-cohort analysis. Again, an association between IBS and alpha diversity was large absent except in one of the 10 cohorts (Figure 5), while it was significant for beta diversity in majority ($n = 6$) of the cohorts (Figure 6). The authors lost this reviewer at Figures 7 and 8, please reconsider if and how to present these two figures. Unlike the case study, the differentially abundant bacteria were not named here, a missed opportunity from both a microbiology (robust microbial markers) and a computational (this analysis vs previous analysis on the same cohort) angle. Lastly, the authors combined all cohorts together for a mega-cohort analysis. This is a great way to increase robustness of their analysis. PCOA plots and differential analysis suggested that the combined dataset had high heterogeneity (Figures 9 and 10), suggesting the necessity for further stratification. While the authors stratified the data via age and cohort exclusion, a more important factor was not considered. That is the subtypes of IBS, including IBS-C (constipation), IBS-D (diarrhea), IBS-M (mixed), which were stratified in most if not all the previous cohorts the authors included here. This is unfortunate in two-folds. One, the authors' own case study already suggested a significant correlation between IBS-D and beta diversity. Two, the mega-cohort is such a great opportunity to bump up the n for each IBS subtypes which are otherwise small in individual cohorts. If publicly available information does not offer a good match between the 16S data and IBS subtypes, the authors should be proactive and contact PIs of the other cohort studies for that information. The PIs are obligated to share that information for their published studies.

In conclusion, the case, cross-cohort, and mega-cohort studies presented here are of high interest to functional GI researchers. However, the cross-cohort and mega-cohort analyses have significant drawbacks in its current form. This reviewer recommends that authors revise their work accordingly to make it a long-lasting piece. To increase transparency and reproducibility, the authors should also deposit their own and reconditioned datasets publicly per ASM's policy, including the case, cross-cohort, and mega-cohort studies.

Specific comments

41, gut bacteria

83, 16S rRNA

84-85, what's the purpose of this step? Please also be more specific - grams of feces per 10ml PBS? filter size? negative control? Please note that 24h of sample processing time is expected to alter the community composition because certain gut bacteria grow very fast.

90, Table 1 is very nice, please link it here and expand it further to list both DNA extraction and sequencing methods for each cohort.

116-118, these are good considerations.

133, reference and rationale using either method?

127, how were archaeal reads handled in both the case and cross-cohort studies? Besides bacteria, 16S sequencing can pick up archaea as well. Of interest, methanogenic archaea have been associated with IBS-C (doi.org/10.1007/s10620-021-06839-0). The Pozuelo cohort even had a specific focus on methanogenic archaea. It will be very interesting if the authors can pull all the archaeal reads from the previous cohorts for a mega-cohort analysis.

150, what IBS symptoms did the 9 without diarrhea have - constipation, bloating, H₂ or CH₄ positive, etc?

159, a high diversity in IBS with diarrhea is unexpected, please discuss further in the context of other studies reported in the literature. Also, a more informative comparison would be HC vs IBS w/ and w/o diarrhea.

166, The ASVs reproduced the same trend as the OTU analysis - that should be the main message here. Please try not to be overly obsessed with the 0.05 p cutoff, the 0.056 p for the D- vs HC is significant enough when considering both ASVs and OTUs.

169-170, partial 16S can't reliably get to the species level - it'd be better to present both OTUs and ASVs instead in figure 4.

Figure 1, briefly explain what each diversity index measures and how it's calculated. All abbreviations should be annotated. Is the P value here FDR adjusted?

Figure 4, Replace the species panel with a OTU panel.

Figure 6, Indicate what the underlined P values mean - why the Saulnier study was not underlined in figure A?

Figure 8, This reviewer has a hard time following the plots here - why not present it the same way as in Figure 5?

Reviewer #2 (Comments for the Author):

Introduction

Line 41: We found that gut bacterias (typo)

Line 58: If IBS is likely to develop/or result in less diversity of gut microbes, how does antibiotic administration improve IBS?

The introduction is very light with minimal information--- I only got one point which is IBS etiology is unknown and related somehow to microbiota structure.

The introduction needs to be more comprehensive, highlights the advances in the field, emphasize the gap, shows novelty of the study design or approach, or expected breakthrough.

Methods

Line 68: diagnosed with IBS (would the author describes how the diagnosis is made just)

Line 72: how the patients got the diagnosis of IBS and there no history of GIT disorder (some patients are as young as 1 year old)

Line 74: two weeks without antibiotics is not enough to restore normal gut flora (6 months at least)

Line 75: why being obese specifically is an exclusion criterion?

Line 77: what do the authors mean by "abnormal endoscopic findings" and why this is an exclusion criterion?

Line 83: fecal samples were frozen at -20 {degree sign}C? for how long-It is advised to ultra-freeze at -80 {degree sign}C but I would assume that has minimal effect when you extract DNA only and not re-culturing the microbes

Line 84: it is not clear how the authors collected the DNA fragments or how they got rid of cell debris and fecal material before DNA extraction---for example, I would use a gradient solution and ultra-centrifuge.

Line 85: vibrated for 24 hours?

Line 86: PowerSoil DNA Isolation Kit is not appropriate for fecal samples-Could the authors justify they choices?

Results

The conclusion is not supported by the results-In what way does this study introduce a rationale for new therapeutic trials?

Suggestion:

A validation study is required

For example:

1) Test the proinflammatory effect of Corynebacteriaceae and Clostridium clostridioforme on GIT cell line such as Caco-2

2) Extract the microbiome cocktail from some stool samples (patients and control) and test the pro or anti-inflammatory effect on cell line---This will help to identify if the gut microbes play a role (as initiation or worsen of the IBS symptoms)

Staff Comments:

Preparing Revision Guidelines

Please return the manuscript within 60 days; if you cannot complete the modification within this time period, please contact me. If you do not wish to modify the manuscript and prefer to submit it to another journal, please notify me of your decision immediately so that the manuscript may be formally withdrawn from consideration by Microbiology Spectrum.

Response to reviewers

Reviewer #1 (Comments for the Author):

IBS is a significant functional GI disorder affecting many millions of people across the globe. What causes IBS remains elusive, but it is believed that gut microbiome may play a significant role. Here, the authors first conducted a small case-control study with a focus on associating IBS with alpha and beta bacterial diversity. It appeared that while the association with alpha diversity was largely absent (Figure 1), it was significant with beta diversity when measured by OTUs (Figure 2). Notably, this significance only remained for the IBS without diarrhea but disappeared for the IBS with diarrhea (IBS-D) subtype. This suggested the IBS patients here maintained an equally rich but distinct microbiome (only IBS without diarrhea) compared to the health controls (HC). The same trend was also observed when measured by ASVs (Figure 3), offering robustness. This distinct microbiome was further illustrated by the differential relative abundance of certain bacterial taxa at the family, genera, species, and ASVs levels (Figure 4), but not the OTU levels. Because partial 16S does not provide species level resolution, the authors should replace the species plot with an OTU plot in Figure 4. Moreover, the authors should have a 4-box plot instead of the 2-box, i.e., IBS D+ vs IBS D- vs IBS vs HC.

→ Thank you for your valuable comments. We modified Figure 4 according to the reviewer's suggestion. We (1) replaced the phrase 'analysis of the species-level aggregated abundance profiles' with 'analysis of OTU abundances', and (2) further stratified the comparison into HC vs. IBS vs. IBS-D vs. IBS-ND. For a given taxon, the figure was accurate in displaying significant difference between HC vs. IBS-ND; however, significant difference was not observed in the case of HC vs. IBS-D. We are grateful to the reviewer's suggestion, as we are now able to better depict the underlying distinctive features of IBS patients with and without diarrhea. Accordingly, the main text was also modified to emphasize how the observed microbial features of IBS-D and IBS-ND overlapped.

Next, the same computational pipeline was applied to a cross-cohort analysis. Again, an association between IBS and alpha diversity was large absent except in one of the 10 cohorts

(Figure 5), while it was significant for beta diversity in majority ($n = 6$) of the cohorts (Figure 6). The authors lost this reviewer at Figures 7 and 8, please reconsider if and how to present these two figures. Unlike the case study, the differentially abundant bacteria were not named here, a missed opportunity from both a microbiology (robust microbial markers) and a computational (this analysis vs previous analysis on the same cohort) angle.

- Figure 7 shows the differential abundance statistics in two dimensions, false discovery rate (i.e., significance) on the Y-axis as well as the magnitude and direction of enrichment (depletion) on the X-axis. Each of the profiled taxa is scattered in the graph, and therefore, we can identify the taxa that are skewed to the left (i.e., less abundant in IBS in this case) or to the right side (i.e., more abundant in IBS). We can also observe taxa with a high significance, i.e., those below the horizontal line demarcating the FDR 0.1 threshold. Hence, the spots (representing taxa) found in the lower-left or lower-right area of the plots represent the robustly identified differential taxa. We acknowledge that such an explanation was not given in the original manuscript, and its inclusion has now strengthened the figure legend for figure 7. We also are in agreement with the reviewer's opinion that the names of differentially abundant taxa would be informative if given here. Hence, we inserted a text box within the figure area to name the taxonomy of the differentially abundant taxa. This label excludes the lengthy list of taxa determined from the Zhu 2019 dataset. We also added a sentence in the main text to mention these taxon names.
- Figure 8 shows the alpha-diversity indices of IBS and healthy controls combined across multiple datasets. In this case, we used a histogram instead of a boxplot (or potentially a violin plot) to more clearly reveal if heterogeneity within samples is a result of merging different datasets. In other words, we wanted to assess if there are multiple peaks within the healthy (or IBS) group that originated from the original study rather than being viewed as a result of an after-effect of combining data sets. Hence we are retaining the original format of the figure. However, to better clarify the nature of the plots, we have revised the figure legend for figure 8. We also suspect that the reviewer might have had trouble understanding what comparison each panel (A-D) represents, and thus, we modified the legend to strengthen and clarify our point.

Lastly, the authors combined all cohorts together for a mega-cohort analysis. This is a great

way to increase robustness of their analysis. PCOA plots and differential analysis suggested that the combined dataset had high heterogeneity (Figures 9 and 10), suggesting the necessity for further stratification. While the authors stratified the data via age and cohort exclusion, a more important factor was not considered. That is the subtypes of IBS, including IBS-C (constipation), IBS-D (diarrhea), IBS-M (mixed), which were stratified in most if not all the previous cohorts the authors included here. This is unfortunate in two-folds. One, the authors' own case study already suggested a significant correlation between IBS-D and beta diversity. Two, the mega-cohort is such a great opportunity to bump up the n for each IBS subtypes which are otherwise small in individual cohorts. If publicly available information does not offer a good match between the 16S data and IBS subtypes, the authors should be proactive and contact PIs of the other cohort studies for that information. The PIs are obligated to share that information for their published studies.

→ We were able to identify the IBS subtype (i.e., IBS-D, IBS-C, IBS-M) for 257 patient samples in total, including 238 adult and 19 pediatric patients. Unfortunately, within the pediatric cohorts, our own cohort was the only one in which the subtype could be resolved. This meant that there was no gain of sample size for pediatric IBS subtypes ultimately. If we were to extend the combined mega-cohort to the subtype-stratified comparisons, we would have to focus on the adult patients. For them, it was possible to compare 124 IBS-D, 66 IBS-C, 48 IBS-M, and 210 HC samples that belonged to the Altomare 2021, Lee 2021, Zhu 2019, and Zhuang 2018 studies. We did not explore the adult IBS subtypes as our story line focuses exclusively on pediatric IBS.

In conclusion, the case, cross-cohort, and mega-cohort studies presented here are of high interest to functional GI researchers. However, the cross-cohort and mega-cohort analyses have significant drawbacks in its current form. This reviewer recommends that authors revise their work accordingly to make it a long-lasting piece. To increase transparency and reproducibility, the authors should also deposit their own and reconditioned datasets publicly per ASM's policy, including the case, cross-cohort, and mega-cohort studies.

→ We added information in the 'availability of data and materials' section as follows:
We have uploaded data files to the zenodo with the DOI 10.5281/zenodo.7272051 (<https://doi.org/10.5281/zenodo.7272051>). Files uploaded include the following: (1) ASV sequences of our samples, (2) ASV read count matrix of our samples, (3) metadata table files, one per each datasets including those from published studies and our own, (4)

combined all-in-one metadata table restricted to IBS vs. control state, (5) OTU read count matrix for each dataset, (6) OTU taxonomy table for each dataset, and (7) species- and genus-level composition matrices for each dataset.

Specific comments

41, gut bacteria → Corrected

83, 16S rRNA → Corrected

84-85, what's the purpose of this step? Please also be more specific - grams of feces per 10ml PBS? filter size? negative control? Please note that 24h of sample processing time is expected to alter the community composition because certain gut bacteria grow very fast.

→ The experimental method was written differently; we have modified it per your comment.

90, Table 1 is very nice; please link it here and expand it further to list both DNA extraction and sequencing methods for each cohort.

→ DNA extraction methods were added to the sequencing method in Table 1.

116-118, these are good considerations. → Thank you. We appreciate the comment.

133, reference and rationale using either method?

→ To be more precise, we used the Bray-Curtis method for the initial fecal sample comparisons, as there was less concern about variation in sequencing depth. Later, we performed cross-cohort comparisons using published datasets. We found that CLR-based Aitchison distance is a more robust method compared to simple rarefaction with respect to depth when dealing with variable sequencing depth. Hence, in the cross-cohort comparisons, we performed a beta-diversity analysis based on the Aitchison distance.

127, how were archaeal reads handled in both the case and cross-cohort studies? Besides bacteria, 16S sequencing can pick up archaea as well. Of interest, methanogenic archaea have been associated with IBS-C (doi.org/10.1007/s10620-021-06839-0). The Pozuelo cohort even had a specific focus on methanogenic archaea. It will be very interesting if the authors can pull all the archaeal reads from the previous cohorts for a mega-cohort analysis.

→ We did not actively filter out the Archaeal reads. Retrospectively, we found that total 19 OTUs from the analyzed datasets were classified as Archaea. Of these, 12 belonged to *Methanobrevibacter* and *Methanosphaera*; the others were unclassifiable below the phylum level. The prevalence of these known human methanogens genera varied greatly among the datasets.

Study	Methanobrevibacter prevalence (% of samples)	Methanosphaera Prevalence (% of samples)
Belkova_2020	60.4	3.8
Pozuelo_2015	26.2	6.9
Lee_2021	15.8	10.5
Saulnier_2011	5.9	0
Zhuang_2018	4.4	0
Mizuno_2017	0	0
Labus_2017	0	0
Altomare_2021	0	0
Zhu_2019	0	0
This study	0	0

Interestingly, the Pozuelo 2015 cohort was the only cohort in which the methanogens were highly common. In contrast, in our cohort, methanogens were not detected in any sample. We have restricted the current study to only the analysis of bacterial compositions, although analyzing the Archae would be very interesting.

150, what IBS symptoms did the 9 without diarrhea have - constipation, bloating, H2 or CH4 positive, etc?

→ The patients had abdominal pain or constipation, and fulfilled the following Rome IV criteria. For at least 2 months before the final IBS diagnosis, they presented the following symptoms:

1. Abdominal pain for at least 4 days per month. This symptom was further associated with one or more of the following: a. defecation, b. a change in defecation frequency, and

- c. a change in the stool form/appearance;
2. In children with constipation, the pain did not resolve with resolution of constipation (children in whom the pain resolves have functional constipation, not irritable bowel syndrome);
3. After appropriate evaluation, the symptoms could not be fully explained by another medical condition.

159, high diversity in IBS with diarrhea is unexpected, please discuss further in the context of other studies reported in the literature. Also, a more informative comparison would be HC vs IBS w/ and w/o diarrhea

- Our case-control study showed no differences in the α -diversity between groups, except for IBS patients with and without diarrhea (Figure 1B). Our case-control study was limited in sample size, and other pediatric cohorts provided no subtype information.
- Based on the adult mega-cohort, we performed a subtype comparison between HC vs IBS with and without diarrhea (Figure 8E). We found that there was a lower α -diversity in patients with IBS with and without diarrhea compared to that of healthy controls. We added them following your recommendation.

166, The ASVs reproduced the same trend as the OTU analysis - that should be the main message here. Please try not to be overly obsessed with the 0.05 p cutoff, the 0.056 p for the D- vs HC is significant enough when considering both ASVs and OTUs.

- Thank you for the comment. We modified this part of the manuscript to emphasize the consistency among the OTU- and ASV-based results.

169-170, partial 16S can't reliably get to the species level - it'd be better to present both OTUs and ASVs instead in figure 4.

- We have addressed this issue as per the reviewer's previous comment.

Figure 1, briefly explain what each diversity index measures and how it's calculated. All

abbreviations should be annotated. Is the P value here FDR adjusted?

→ Missing annotations for abbreviations were added to the figure legend. The *P*-value indicated here is not an FDR-adjusted version, and we modified the legend to clarify this.

Figure 4, Replace the species panel with a OTU panel.

→ Figure 4 was updated to have an OTU panel instead of a species panel. The issue with Figure 4 is addressed in more detail in one of the reviewer's previous comments.

Figure 6, Indicate what the underlined P values mean - why the Saulnier study was not underlined in figure A?

→ *P*-values below 0.05 are underlined. Those for Saulnier's study were not underlined by mistake. We have rectified the figure and explained in the legend as to why some *P*-values are underlined.

Figure 8, This reviewer has a hard time following the plots here - why not present it the same way as in Figure 5?

→ We addressed this issue with one of the comments from another reviewer. We are adding the same response here: Figure 8 shows the alpha-diversity indices of IBS and healthy controls combined across multiple datasets. In this case, we used a histogram instead of a boxplot (or potentially a violin plot) to more clearly reveal if heterogeneity within samples results from merging different datasets. In other words, we wanted to assess if there are multiple peaks within the healthy (or IBS) group that originated from the original study rather than being viewed as a result of an after-effect of combining data sets. Hence we are retaining the original format of the figure. However, to better clarify the nature of the plots, we have revised the figure legend for figure 8. We also suspect that the reviewer might have had trouble understanding what comparison each panel (A-D) represent and thus we modified the legend to strengthen and clarify our point.

Reviewer #2 (Comments for the Author):

Introduction

Line 41: We found that gut bacterias (typo)

→ Corrected

Line 58: If IBS is likely to develop/or result in less diversity of gut microbes, how does antibiotic administration improve IBS?

→ Small intestinal bacterial overgrowth (SIBO) may coexist with irritable bowel syndrome (IBS) in nearly half of the patients. Thus, eradication therapy has been reported as effective in reducing IBS symptoms.

The introduction is very light with minimal information--- I only got one point which is IBS etiology is unknown and related somehow to microbiota structure.

The introduction needs to be more comprehensive, highlights the advances in the field, emphasize the gap, shows novelty of the study design or approach, or expected breakthrough.

→ We appreciate your comment. We have now added the novelty of our study as per your suggestion.

Methods

Line 68: diagnosed with IBS (would the author describes how the diagnosis is made just)

→ We have cited a reference for the diagnostic criteria for IBS and have added details of the classification used in this study.

Line 72: how the patients get the diagnosis of IBS and there no history of GIT disorder (some patients are as young as 1 year old)

→ The patients had abdominal pain or constipation and fulfilled the following Rome IV criteria. For at least 2 months before the final IBS diagnosis, they presented the following symptoms:

1. Abdominal pain for at least 4 days per month. This symptom was further associated with one or more of the following: a. defecation, b. a change in defecation frequency, and c. a change in the stool form/appearance;
2. In children with constipation, the pain did not resolve with resolution of constipation (children in whom the pain resolves have functional constipation, not irritable bowel

syndrome);

3. After appropriate evaluation, the symptoms could not be fully explained by another medical condition.

The diagnosis of IBS was made with clinical histories using ROME IV. The final diagnosis implies no history of GIT disorders, such as inflammatory bowel disease, allergic GI diseases, or intestinal failure. Therefore, we removed ‘No history of GIT disorder’ to reduce confusion. The inclusion criterion was children aged 4 – 18 years diagnosed by the ROME IV criteria for children/adolescents.

Line 74: two weeks without antibiotics is not enough to restore normal gut flora (6 months at least)

→ Thank you for your comment. We recruited children who were not administered any antibiotics for at least two weeks prior to enrollment (visit 1). We collected their stools 2 ~ 4 weeks after the first visit (visit 2). Therefore, the wash-out period without antibiotics was 4~6 weeks. Other studies had one month wash-out period without antibiotics (Tap 2017 *Gastroenterol* <https://doi.org/10.1053/j.gastro.2016.09.0491>; Vervier 2022 *Gut* <https://doi.org/10.1136/gutjnl-2021-3251771>), while some waited two months after the antibiotics were stopped (Gobert 2016 *Sci Rep* <https://doi.org/10.1038/srep393992>). Some studies reported that the recovery periods depend on the types of antibiotics. For e.g., ampicillin (commonly used antibiotics in children) needs one month, vs. vancomycin and meropenem (not used in our patients for six months). We modified the sentences correctly and added relevant limitations.

Line 75: why being obese specifically is an exclusion criterion?

→ Previous studies have shown that gut microbiota in obese children differs from healthy controls. Therefore, we excluded obese children to reduce bias. We also collected their medical histories; no children had chronic diseases or obesity histories.

Line 77: what do the authors mean by "abnormal endoscopic findings" and why this is an exclusion criterion?

→ The inclusion criterion of IBS implies that no other accompanying GIT disorders exist. Patients with abnormal endoscopic findings, such as inflammation in bowel mucosae, are not diagnosed with IBS. We thus deleted the sentence and have explained the detailed diagnostic process. In clinical practices, colonoscopies are performed in patients with high calprotectin and suspected symptoms or signs of inflammatory bowel disease. In this study, one (fecal calprotectin level of 268 mg/kg) out of 19 patients underwent colonoscopy and gastroduodenoscopy. The findings were normal and thus the patient was diagnosed with IBS. Two patients were excluded because they showed high fecal calprotectin levels, and endoscopic biopsy revealed eosinophilic gastroenteritis, not IBS.

Line 83: fecal samples were frozen at -20 {degree sign}C? for how long-It is advised to ultra-freeze at -80 {degree sign}C but I would assume that has minimal effect when you extract DNA only and not re-culturing the microbes

→ Although there is not enough literature on this subject, Gavriliuc et al. 2021 (<https://doi.org/10.7717%2Fpeerj.10837>) demonstrated that storing fecal samples (horse feces in their case) for multiple months to years at -20 degree does not alter the bacterial community composition as compared to storage at -80 degree.

Line 84: it is not clear how the authors collected the DNA fragments or how they got rid of cell debris and fecal material before DNA extraction----for example, I would use a gradient solution and ultra-centrifuge.

Line 85: vibrated for 24 hours?

→ The experimental method was written differently and now we have modified it.

Line 86: PowerSoil DNA Isolation Kit is not appropriate for fecal samples-Could the authors justify their choices?

→ PowerSoil DNA kit is undoubtedly the most popular and established choice for human fecal/stool metagenome sequencing studies. Its performance has been validated in the literature too (doi.org/10.1038/s41598-019-49520-3).

Results

The conclusion is not supported by the results-In what way does this study introduce a rationale for new therapeutic trials?

→ We agree with your comment and have corrected the conclusion as follows:

To our knowledge, we performed the first cross-cohort analysis to find the association between IBS and gut microbial diversity and composition. It revealed that gut bacterial dysbiosis is associated with IBS, but the causal relationship is uncertain. Further studies are needed to ascertain whether the change in intestinal microorganisms contributes to developing IBS.

Suggestion:

A validation study is required

For example:

- 1) Test the proinflammatory effect of Corynebacteriaceae and Clostridium clostridioforme on GIT cell lines such as Caco-2
- 2) Extract the microbiome cocktail from some stool samples (patients and control) and test the pro or anti-inflammatory effect on cell line---This will help to identify if the gut microbes play a role (as initiation or worsen of the IBS symptoms)

→ We appreciate your valuable advice. Future research should include validation studies as you comment rightly suggests.

December 10, 2022

Prof. Jung Ok Shim
Korea University College of Medicine
148, Gurodong-ro, Guro-gu
Seoul 08308
Korea (South), Republic of

Re: Spectrum02125-22R1 (Gut bacterial dysbiosis in irritable bowel syndrome; a case-control study and a cross-cohort analysis of publicly available datasets)

Dear Prof. Jung Ok Shim:

It's my pleasure to inform you that I have decided to accept your manuscript for publication in Microbiology Spectrum. Thank you for the efforts in addressing reviewer and editor concerns in your revised manuscript. You and your team have done an excellent job responding to all of our concerns, and the manuscript is substantially better as a result. I'm also particularly appreciative of the added sharing of data, and the expanded details provided in the materials and methods. Overall, I now feel it reads at the "level" of the potential impact of this work. Thus, I have also suggested to ASM's press office that this manuscript might be considered for highlighting in the press once published - but please understand I can't guarantee this.

Next, I am forwarding it to the ASM Journals Department for publication. You will be notified when your proofs are ready to be viewed.

Sincerely,

Jonathan Jacobs
Editor, Microbiology Spectrum
